# Real-Time Advanced Computational Intelligence for Deep Fake Video Detection

Nency Bansal [1], Turki Aljrees [2,*], Dhirendra Prasad Yadav [3], Kamred Udham Singh [4], Ankit Kumar [3], Gyanendra Kumar Verma [1] and Teekam Singh [5]

1 Department of Computer Engineering, National Institute of Technology, Kurukshetra 136119, India
2 Department College of Computer Science and Engineering, University of Hafr Al-Batin, Hafar Al-Batin 39524, Saudi Arabia
3 Department of Computer Engineering & Applications, GLA University, Mathura 281406, India
4 Department School of Computing, Graphic Era Hill University, Dehradun 248002, India
5 Department of Computer Science and Engineering, Graphic Era Deemed to be University Dehradun, Uttarakhand 248002, India
* Correspondence: tajrees@uhb.edu.sa

**Abstract:** As digitization is increasing, threats to our data are also increasing at a faster pace. Generating fake videos does not require any particular type of knowledge, hardware, memory, or any computational device; however, its detection is challenging. Several methods in the past have solved the issue, but computation costs are still high and a highly efficient model has yet to be developed. Therefore, we proposed a new model architecture known as DFN (Deep Fake Network), which has the basic blocks of mobNet, a linear stack of separable convolution, max-pooling layers with Swish as an activation function, and XGBoost as a classifier to detect deepfake videos. The proposed model is more accurate compared to Xception, Efficient Net, and other state-of-the-art models. The DFN performance was tested on a DFDC (Deep Fake Detection Challenge) dataset. The proposed method achieved an accuracy of 93.28% and a precision of 91.03% with this dataset. In addition, training and validation loss was 0.14 and 0.17, respectively. Furthermore, we have taken care of all types of facial manipulations, making the model more robust, generalized, and lightweight, with the ability to detect all types of facial manipulations in videos.

**Keywords:** fake news; XGBoost; mobNet; Efficient Net; deep fake network; deep fake detection challenge

## 1. Introduction

Deep learning techniques are nowadays being exploited to create fake videos. These fake videos are often created with malicious intent, like defaming a famous political leader, fake pornography of famous actors/actresses, tampering with evidence related to forensics and courts, and creating scams and frauds for identity manipulation. Deep learning techniques are used for these purposes, as the fake videos generated from these techniques look very similar to the original videos. It is not even possible for humans to differentiate between real and fake images/videos. The deep learning technique used to generate such videos is known as a Generative Adversarial Network (GAN) [1].

DeepFake is made up of two words, "Deep" and "Fake". It means creating a fake image or video using deep learning techniques. Deep learning techniques were designed to create videos that appear similar to the original videos using facial manipulation or face swap techniques. Such videos are designed for various gaming purposes and are commonly used in graphic design and animation. Facial manipulation has a broad scope in the movie industry for creating animations, etc. Sometimes, fake images/videos are also created for recreational purposes. These techniques have been used and implemented for many years, generally only for legal purposes. Other than deep learning techniques, there are various mobile apps such as FaceApp, Zao, FaceSwap [2], Reface, etc., easily available

on the Play Store, which allows creating such fake videos in a free-of-cost and easy way, as mentioned in [3]. A sample of fake images created from such apps is shown in Figure 1. If shown randomly, it is difficult to differentiate between the real and fake images. The fake images can be easily generated by anyone, irrespective of age or knowledge. deepfake detection has become an emerging area of research due to the above reasons. Moreover, it is a challenging and unsolved problem to date.

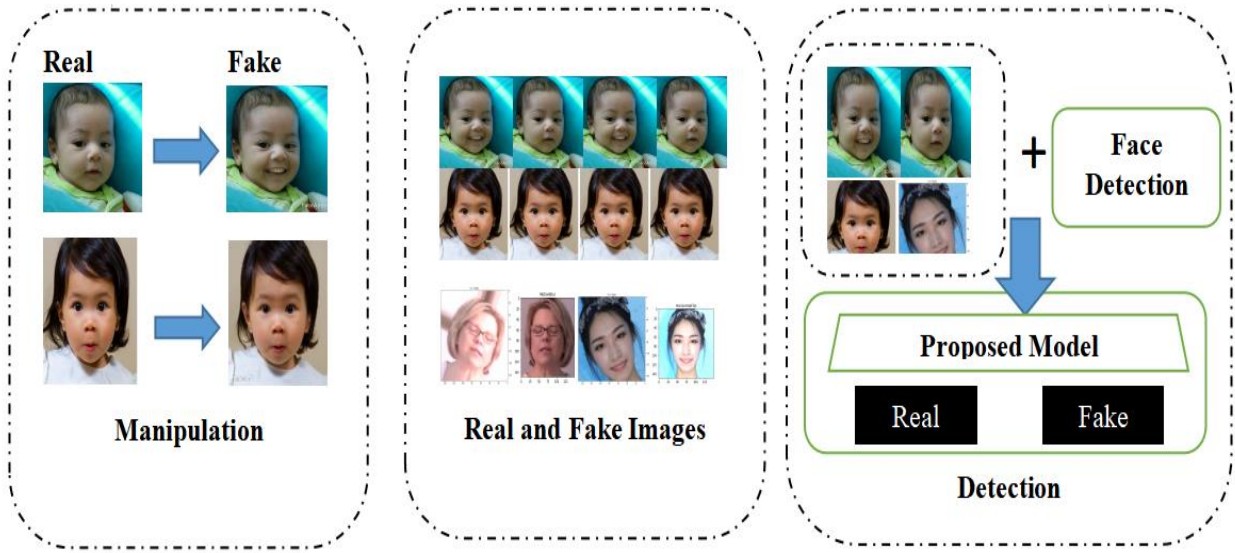

**Figure 1.** High Level view of deepfake creation and detection model.

Deepfake allows for the creation of fake videos similar to realistic videos by swapping the complete face of one person with another, or sometimes just by manipulating the eye movement, lip movement, and expression to simulate some other context. In 2018, a fake video of Barack Obama (former president of the US) was released by BuzzFeed in which he was talking about some subject [4]. That deepfake video was created using FakeApp [5] software. It raised many concerns about spreading the wrong information on social media, impersonation, identity theft, etc. It is becoming more threatening when fake videos of our leaders are created to spread rumors, to spread violence, and other falsification purposes [6,7]. It can also be used to create fake satellite images of Earth, to create objects which are not at all present in reality, which can be used by attackers to mislead whole troops of military and analysts, etc., to misguide them to a place where they can be easily attacked or killed [8].

The number of deepfake videos doubles every eight months, according to available statistics. Detecting and removing fake data on the internet can prevent misinformation, rumors, and fake pornography from spreading if we cannot prevent its creation [9]. To achieve this, our detection methods must be robust, generalized, fast, and accurate enough to detect fake data and punish those who are generating and misusing such data. The ResNet 152 [10] and Xception [11] models are performing well in the field of object recognition, as reported in the literature. In addition, several deep learning methods have been successfully applied in several other domains [12–15]. The proposed model was trained on a large DFDC dataset [16] with a motive to create a robust, generalized, and more accurate model. The DFDC dataset has no data leaks, and our results are not dependent on a particular feature of a face like teeth, smile, eye, lip, etc. We considered all the features of a face, including a complete face swap. We also implemented various image augmentation techniques while training our model to improve the performance gains.

Figure 1 shows the abstract view of the deepfake creation and detection model. The first block shows the creation of fake images, manipulated using various techniques. The second block shows the set of real and fake images. It can be considered a dataset, which contains both real as well as fake images. The last block shows the compact view of the

detection process. Frames are extracted from videos, and each face is detected from the frames. Then these faces are provided as input to the detection model, where predictions of real and fake videos are made. The main contributions of this work are as follows.

i.     We propose a new model architecture, which consists of a linear stack of separable convolution 2D, max-pooling layers with XGBoost as the classifier and modification Swish as an activation function.

ii.     Rather than focusing on any single facial manipulation technique, we focused on generating a robust, scalable, and generalizable model for deepfake video detection by training the model on an augmented and generalized dataset.

iii.     The proposed model outperforms on deep fake dataset with less training and validation loss.

The rest of this paper is organized as follows. Sections 2 and 3 detail related work and preliminaries of CNN (Convolution Neural Network). Section 4 introduces the proposed model. Section 5 presents experimental details and results of the proposed architecture on the DFDC dataset. Finally, Section 6 gives the conclusions and future scope.

## 2. Literature Review

This section provides a literature study based on the merit and demerit of the methods used by various researchers. Aya et al. [17] used YOLO-CNN-XGBOOST, where YOLO (you only look once) is the face detector used to extract the faces from the video frames, InceptionResNetV2 CNN is used to extract the features from the detected faces, and XGBoost is used as the classifier on the top of the CNN network to classify the video as fake or real. They used the CelebDF-FaceForencics++ merged dataset for training and testing their model and attained an accuracy of 90.73%. Their dataset consists of 2848 training and 518 testing data selected from CelebDF and FaceForensics++. Their dataset is not large enough to test and train a deep neural network. Hsu et al. [18] implemented a two-phase deep learning technique for the detection of deepfake images. In the first phase, they extracted features based on the popular CFFN (Common Fake Feature Network). They used the Siamese network architecture presented in [19]. CFFN uses multiple units in which each unit has multiple blocks that are used to increase the representative capability of the fake images. Discriminative features between the real image and fake images are extracted as part of the learning process of the model. Then, the features are fed as inputs to the second phase, which is CNN combined with the last layer of CFFN. The accuracy claimed by the researchers is 90.04%.

Tolosana et al. [20] mainly focused on four types of face manipulation techniques. (i) Entire Face Synthesis: In this method, the complete face of the target image replaces the complete face of the source image. (ii) Attribute Manipulation: In this technique, some attributes like hairs, specs, ageing, etc., are replaced in the source image by that of the target image. (iii) Expression Swap: in this technique, expressions of the image like a smile or eye features are manipulated to match with those of the target image. (iv) Identity Swap: In this technique, the identity of the source image is replaced with that of the target image. They experimented on already existing models, which worked well on the seen data; however, a model was not generalized for complex data. They also mentioned that although the models perform well on specific datasets, like UADFV and FaceForensics++ [21], their accuracy suffers with much error when applied to a generalized dataset like the DFDC dataset.

Mirsky et al. [22] defined deepfake as "Believable media that is created by a branch of machine learning (Deep Neural Network)". They have categorized human visuals into Reenactment and Replacement categories. In Reenactment, a few attributes are manipulated, like the pose of a person, head position, mouth position for facial expression, gaze of a person that includes eye position, eyelid position, etc. In Replacement, the complete face is swapped or transposed from the source image to the target image. They also focused on various technology drawbacks related to deepfake, and the current status of attacker and defender games, to give a deep insight to the readers and guide them in future research.

Güera and Delp [23] used a RNN (Recurrent Neural Network) model for deepfake video detection. According to the researchers, a simple pipeline architecture can work well and can achieve competitive results. They used LSTM for this purpose. As LSTM is designed for sequential data, this model does not work well on unseen data. Moreover, this model also takes much time to converge for high-resolution images, and thus a long time to predict the results.

In [4], Suwajanakorn et al. described the procedure for creating a fake video. They proved that after training a model for a large dataset, the model maps the lip shape and lip movement as per the change in audio. They showed lip-sync merged with the new audio with such perfection that it was difficult for the human eye to recognize that it is a fake video. They created a fake video of Barack Obama (former President of the USA). All the lip shapes, lip movement, mouth shapes, texture, and emotional changes were taken care of to maintain sync with the audio. Their pipeline consisted of a manual step to select and mask a teeth proxy as per the target video, which can be automated in future work. The method that they used relies mostly on MFCC audio features. Most people have conducted research based on various facial features to detect whether the video is fake.

Jung et al. [24] developed a method to analyze a spontaneous, unconscious, and natural human function, that is, eye blinking. According to them, by analyzing the eye blinking pattern, we can detect whether a video is real or fake, as eye blinking is natural, and it is very difficult to make it fake. This is a type of data leak, and this model does not give good accuracy in a generalized dataset. This model performs well on a particular dataset which has input related to eye blinking and hence does not works well on generalized data. They proved their accuracy only on a particular dataset. They classified a video as fake if blinking eyes were not found in consecutive frames of a video.

In thepaper [25], proposed methods for deepfake detection which do not require much training data, hence saving resources and time. In the researchers explained various techniques used to create deepfakes and the techniques used to detect them. Creating fake videos and images has become a very easy task nowadays. No special knowledge or experience is required to create such images or videos. Much software is available free of cost to create such videos and images. Detection of such images and videos has become a challenge. They surveyed many detection techniques in which deep learning techniques perform well on the generalized dataset.

Sergey et al. [26] proposed an optical flow-based technique for deepfake video detection. Instead of finding features in a single frame, they tried to exploit dissimilarities between the frames. For that, they used the optical flow technique. This technique was then used as a feature to be learned by various CNN classifiers. This technique can be used with other state-of-the-art models to further improve the results. After analyzing this technique, we found that it will only be able to detect some types of fake videos. This technique performs well on the training data set and only those types of manipulations on which the model is trained. Thus, this technique does not fit in a real-time environment.

Recently, Vamsi et al. [25] applied CNN and Long Short-Term Memory (LSTM) for fake video detection and achieved a classification accuracy of 91%. However, the computation cost of the model is high due to a large number of trainable parameters. They have also studied several other techniques and models developed for Deepfake detection, and concluded DeepFake is still a challenge to the social community that should be resolved. They also provided a path for researchers to develop a generalized and robust model.

A good model must not be dependent on a dataset or a feature to detect the fake video. To the best of our knowledge and study, we have not found a model which performs well on unseen data, is able to detect fake videos in a real-time environment, and is computationally efficient. Thus, we developed a generalized, robust, and efficient method to solve the global challenge.

## 3. Preliminaries

### 3.1. Generative Adversarial Network (Gan)

As GAN [27] models are continuously evolving, verifying the integrity and authenticity of deepfakes has become more difficult. Earlier methods used to verify the integrity have lost their effectiveness due to the rapid evolution of such features and the loop-holes exploited by the attackers. Therefore, we mainly focused on finding new elements and features for our study's integrity verification and deepfake detection. We referred to all recent surveys to collect the overall background knowledge of the topic, to know what work has already been done, what accuracy has been achieved, and what could be the new elements that we can contribute in this area of research to improve the performance and to create a more generalized model.

### 3.2. Efficient Net Model

Convolutional Neural Networks (CNNs) work on recurrent learning methods. In the training phase, a CNN randomly assigns weights to all the features, finds the results, and matches them with the actual results to calculate the error. Based on the error, it reassigns or adjusts the weights of all the features. This is continued to obtain good accuracy. Due to this feature, people have thought of increasing the number of layers in the convolutional neural network to achieve higher accuracy. Dropout is used at each layer to train different neurons on different features.

If all the neurons are trained on the same feature, they will all give the same result for an unseen image. This reduces the accuracy of the model. To avoid this situation, the dropout is tuned as per the requirement. By increasing the number of layers, accuracy is improved to a great extent. Many experiments were performed to find the saturation level of the number of layers of a convolution neural network. It reached a saturation level on 152 layers in the ResNet152 model. By increasing the number of layers, we increase the depth and, thereby, the complexity of the model. As the complexity increases, the time taken to train the model also increases. This leads to the requirement of high computational resources like GPU, memory, SSD, etc.

To reduce the complexity and the need for computational resources, researchers found a solution in terms of the horizontal scaling of the model. In 2019, Tan and Le proposed a new model, created by scaling up the convolutional neural network on all three aspects of width, depth, and resolution. This model was named the Efficient Net Model [28].

They determined that although it is very critical to balance all three aspects (width, depth, and resolution) in a particular model, if we scale up our model on all three aspects while maintaining balance among all three aspects, we can achieve high accuracy with less computational resources. This model is $8.4\times$ smaller and $6.1\times$ faster on inference than the best existing convolutional neural networks.

They proved their high accuracy against various other existing models. Through extensive research, they derived an optimal formula with the following coefficients to maintain a balance among all three aspects: Depth = 1.20, Width = 1.10, Resolution = 1.15. This means that to scale up the CNN Model, the depth of layers must increase by 20%, width by 10%, and resolution by 15% to achieve the highest efficiency possible while expanding the implementation and improving the accuracy of the CNN model. This model is designed for the segmentation of images (Table 1). The original output layer consists of 1000 outputs. We added a linearly fully connected layer with the ReLu activation function to get the binary output. Two GPUs were required for training.

**Table 1.** Parameters used in the Efficient Net model.

| Parameters | Used Value |
|---|---|
| Drop Out | 0.65 |
| Learning Rate | 0.001 |
| Epochs | 15 |
| Batch Size | 64 |
| Optimizer | Adam |

The Efficient Net model uses the method of compound scaling, which uses a compound coefficient $\varphi$ for scaling all the parameters like network width, depth, and resolution uniformly in a principled way:

$$\text{Depth}: \ d = \alpha^{\phi} \tag{1}$$

$$\text{Width}: \ w = \beta^{\phi} \tag{2}$$

$$\text{Resolution}: \ r = \gamma^{\phi} \tag{3}$$

$$s.t. \alpha * \beta^2 * \gamma^2 \approx 2$$

$$\alpha \geq 1, \beta \geq 1, \gamma \geq 1$$

where $\alpha$, $\beta$, and $\gamma$ are constants that can be determined by a small grid search. Intuitively, $\phi$ is a user-specified coefficient. They used this in different versions of the Efficient model from B0 to B7. This has increased the complexity of the model. Hence B7 is the highest complexity model and B0 is the minimum complexity version of the model.

### 3.3. Xception Model

The Xception model [29], developed in 2017 by Francois Chollet, is an interpretation of the Inception model. It is inspired by "Network in Network Architecture". It is a very lightweight model. The training time of this model is less compared to other models. It has used 22 million parameters. Features decrease as we go down the line in this model. ReLu is used as the activation function and batch normalization in all the layers. There are 24 players in the middle flow of the model architecture for feature extraction, making the model less complex. The goal of this model was to act as a "Multi-Level Feature Extractor". It is also known as "Extreme Inception".

It has replaced Inception modules with depth-wise separable convolutions, which has helped improve gains on different datasets. It has shown the difference between the Inception module and depth-wise separable convolutions. It has 36 layers. These 36 layers are divided into 14 modules that are linearly connected, except the first and the last module. It can also be seen as a linear stack of depth-wise separable convolutional layers with residual connections. We used a pre-trained Xception model [30] for image classification. We tuned its hyper parameters to improve the performance gains on our dataset. We experimented with various parameter combinations and concluded that the following parameters give the best accuracy among all available options shown in Table 2.

**Table 2.** Parameters Used in the Exception Model.

| Parameters | Used Value |
|---|---|
| Drop Out | 0.75 |
| Learning Rate | 0.001 |
| Epochs | 20 |
| Batch Size | 128 |
| Optimizer | Adam |

## 4. Proposed Methodology

The biggest contribution of this work is to propose a new model architecture for deepfake video detection, which is a precise and optimized combination of two state-of-



the-art models: Xception and Efficient Net. As described above, both models have their strengths and weaknesses in terms of horizontal and vertical computation of the models, the complexity of the models, the number of layers in both models, etc. We first worked on each model separately and then optimized both models by making required changes like parameter values to get the maximum performance gains.

We created our new model architecture by taking into consideration the properties of both models and combining them in an efficient way to get high performance with a marginal increase in computational complexity. Apart from creating a new architecture, we also worked on our dataset for better performance of the model in a real-time environment. As the quality of the dataset has a large impact on the performance of the model, we worked to improve the quality of the dataset. We applied various image argumentation techniques described below while training the model.

### 4.1. Image Augmentation

Image augmentation is used to improve size of the dataset for the better training of the CNN model so that overfitting of the model can be avoided. In this study, Shift Scale Rotate, Horizontal Flip, Normalize, Random Brightness Contrast, Motion Blur, Blur, Gauss Noise, and JPEG Compression techniques were used for data augmentation. Implementing these techniques with optimized parameters has led to 0.3% performance gains. Moreover, these techniques teach the model with various types of inputs, thereby helping in making the model more robust and accurate.

Figure 2 illustrates various augmentation techniques applied on the images during training of the model.

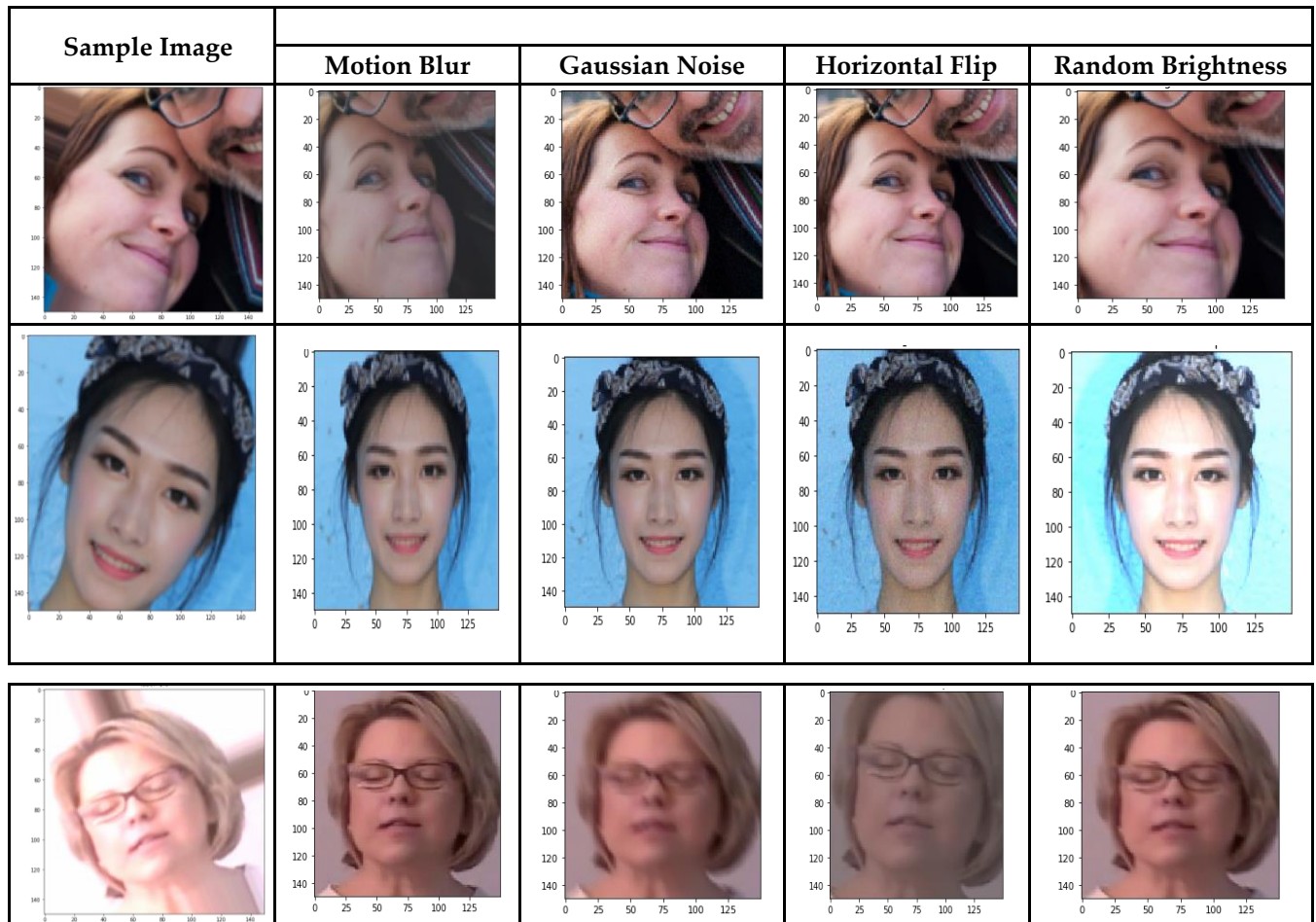

**Figure 2.** Examples of image augmentation.

### 4.2. Construction of Fully Connected Output Layers

Fully connected layers were applied to the original output of our proposed model. Fully connected layers include the following components, as shown in Figure 3. First, we flattened the output using the flatten method of Pytorch. Then dropout was used. After that, a linear layer for reducing the dimensions of the output was applied. Finally, the ReLu activation function was used.

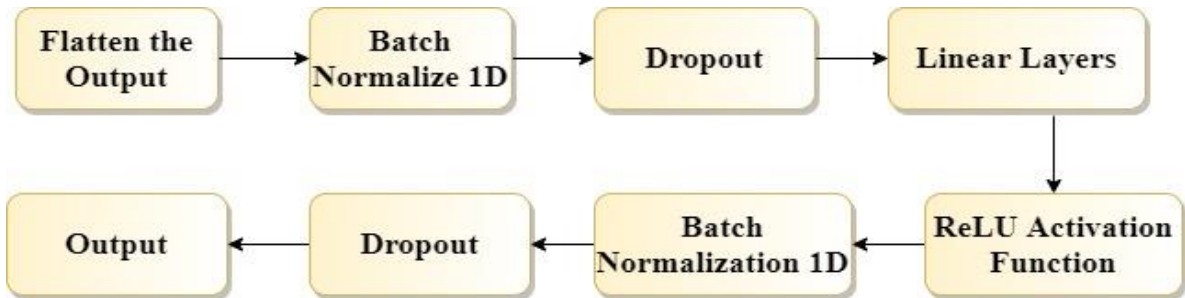

**Figure 3.** Flow chart of fully connected layers of the proposed model.

The ReLu (Rectified Linear Activation Function) can accelerate the training process on small positive or negative weights. In addition, it reduces the range of the output. The batch normalization 1D was applied. This makes the deep CNN faster and more stable by normalizing the input layers for each mini-batch through rescaling and refactoring. Further, it reduces the number of epochs required to train the model.

### 4.3. Architecture of the Proposed DeepFake Detection Model

In our proposed model architecture, we relied on the strength of the Xception model. While training, we observed that Xception is a swift model, and it gets trained very fast. When we went through the Efficient Net architecture, we found that the basic block of the Efficient Net model is the MB convolution block. The original MB convolution block consists of a separable convolution layer 2D with ReLu activation function, followed by a depth-wise convolution layer 2D with ReLu activation function, followed by a global averaging pooling layer, followed by a convolution 2D layer for squeeze and expansion.

We created the DFN, which consists of 48 layers. These layers are divided into 17 modules. In addition, the architecture is divided into three components: entry flow, middle flow, and exit flow. In the entry flow, the max-pooling layer plays a vital role. The basic block of entry flow is repeated thrice, preceded by two standard convolution layers. Each has a stride of two, which means that the kernel takes the hop of two rows and columns when convolution is performed. In the middle flow, the basic building block is repeated eight times. The output of one block is treated as input when the same block is repeated the next time. In this middle flow, we added three MB convolution blocks which make our architecture more accurate and help in better feature extraction. Further, we added three MB Convolutions in the middle flow of our architecture based upon the experiments performed. Our motive is to create a robust and generalized model architecture without increasing the computational complexity of the model.

In exit flow, the basic building block is followed by a global pooling layer and a fully connected layer. The main use and advantage of using a fully connected layer are explained in Section 3.2. Finally, the output of all these layers is passed as an input to the classifier, which finally classifies the image. We used XGBoost as the classifier to classify the image into various classes. First, weights of the Image-Net are used as the initial weights in our proposed model rather than the random weights, which makes our model converge faster and improves its performance. Our model has depth-wise convolution instead of normal convolutions to reduce the computation cost. The numbers of multiplication operations are more in the standard convolution layer; hence it is more computationally expensive

and time-consuming. In standard convolution, filters across all input channels and the combination of these values are made in a single step.

In contrast, in depth-wise separable convolution, this step is performed in two steps. The first step is depth-wise convolution, which it performs in the filtering stage. It applies convolution to a single input channel, whereas, in standard convolution, convolution is applied to all the channels simultaneously. The convolution operation is nothing but element-wise multiplication and adding them all. The second step is point-wise convolution which performs the combining operation. Point-wise convolution involves the linear combination of each output of the layers. Depth-wise separable convolution is much cheaper (in terms of multiplications) and is computationally efficient. The input image is passed through all the layers, and depth-wise separable convolution operations are applied. Max pooling and global average pooling layers reduce the dimensions and train the model in less time. The two-step method has been used to reduce the multiplication operations and make the model computationally efficient.

MB Convolution block has three components. Each of these blocks has residual connections from start to finish. Supposing the first layer has found some good features and there are many more layers ahead of it, all the layers need to enrich those features, extract new features, and carry already found features. The residual connections are important from the first layer to the classifier to carry features found in earlier layers. In the MB Convolution block, $1 \times 1$ convolution is applied on input. This convolution is applied to expand the input to a much high dimensional set. Then, on that set, depth-wise convolution is applied. Then again, $1 \times 1$ convolution is applied to the output of depth-wise convolution. This time it is used to convert the data back into its original dimension, the same as the input. Then all such outputs are added together. This whole process is also known as the expand and squeeze process.

Feature creation is considered a two-step process. One is feature aggregation, which is to group similar features, and in step two, each group is processed separately to create new features. First, $1 \times 1$ convolution is considered feature aggregation, and depth-wise convolution is considered a feature creation step. The $1 \times 1$ convolution is much faster and cheaper as it saves lots of multiplication operations. Depth-wise convolution is used to overcome the complexity created by the increasing depth of the model. Each of the MB convolution blocks enriches the feature set by projecting it to the high-dimensional block where feature aggregation and creation are performed. Hence, even after adding the layers, our model is computationally efficient and fast.

Further, we experimented and verified that the Swish activation function performs better than the ReLu activation function, and we gained 0.7 performance gains by replacing ReLu with Swish. Thus, we used the Swish activation function after all the layers in our proposed architecture. Each convolution layer, separable convolution layer, and depth-wise convolution layer is followed by batch normalization [17].

We used a depth-wise convolution layer as it considers depth dimensions with spatial dimensions like width and height. An input image has three channels for the interpretation of redness, greenness, and blueness of the pixels. However, after a few convolutions, the number of channels increases. Each channel can be interpreted as the interpretation of an image.

We experimented with a different number of MB convolution blocks. However, we used three MB convolution blocks to aim at the minimum complexity of the model. These blocks are added to the middle layer of our architecture for feature extraction. All convolution layers use a depth multiplier of one for no depth expansion. There are 25 million parameters, which is a bit more than that of the Xception model, but very much less than the Efficient Net model.

Our proposed model architecture maintains a trade-off between the complexity of the model, including the training time of the model, prediction time for an input video, computational complexity, resources required for training our model, etc., and performance of the model on the DFDC dataset for deepfake video detection. The proposed model

architecture is still a lightweight model, with less computation complexity and higher performance gains. In short, the proposed model architecture is a linear stack of depthwise convolution layers with residual connection except in the first and the last layer and the inclusion of MB blocks from Efficient Net architecture. Other parameters, like the Swish activation function, global average pooling layer, and a fully connected layer at the output, are added for better performance of the model in terms of both computational time complexity and accuracy discoed in Algorithm 1.

---

**Algorithm 1:** Algorithm for DeepFake Video Detection

---

**Result: Real/Fake Video**
Read a video from the dataset.
Divide the video into frames using OpenCV and label all the image frames as fake or real using one-to-one mapping.
Face is detected from these frames using the BlazeFace [31,32] library.
Apply image augmentation techniques while training the model.
Provide frames with faces as input to our model for classification.
Our model classifies the image as real (1) or fake (0).

---

If any frame of a video is tagged or labeled as fake, then that video is declared as fake, and if all the frames of a particular video are labeled as real, then that video is labeled as real. On this labeling basis, videos are classified as real or fake by our model.

Figure 4 shows the architecture of our proposed model. We divided our architecture into three parts, namely Entry Flow, Middle Flow, and Exit Flow. Data enters the entry flow, passes through all the layers of each block, enters the first block of the middle layer which gets repeated eight times, then enters the second block of the middle layer which gets repeated three times. The middle layer is also known as the feature extractor layer. The output of this layer is loaded into Exit Flow, from where we get the final output.

Figure 5 shows the working of our model without many technical details. This shows how various steps are performed during the training and testing of the model.

### 4.4. Computational Complexity of Our Model

The computational complexity of a model can be understood as the resources required by the model for its complete execution. The resources required include the system or the memory requirements and the needs for time required by the model to converge and produce the required results. Our model works on video datasets. As a result, it requires a substantial amount of space. Additionally, the model has 48 layers, which contributes to its high computational complexity. Despite this, we simplified our model by using depth-wise separable convolutions rather than regular convolutions. As a result, we are able to save time and reduce the number of times multiplication is required. Compared to other state-of-the-art models such as Xception and Efficient Net, our model requires more training time. The performance of our model is superior to that of other models. Increasing the marginal complexity of our system allows us to achieve high-performance gains.

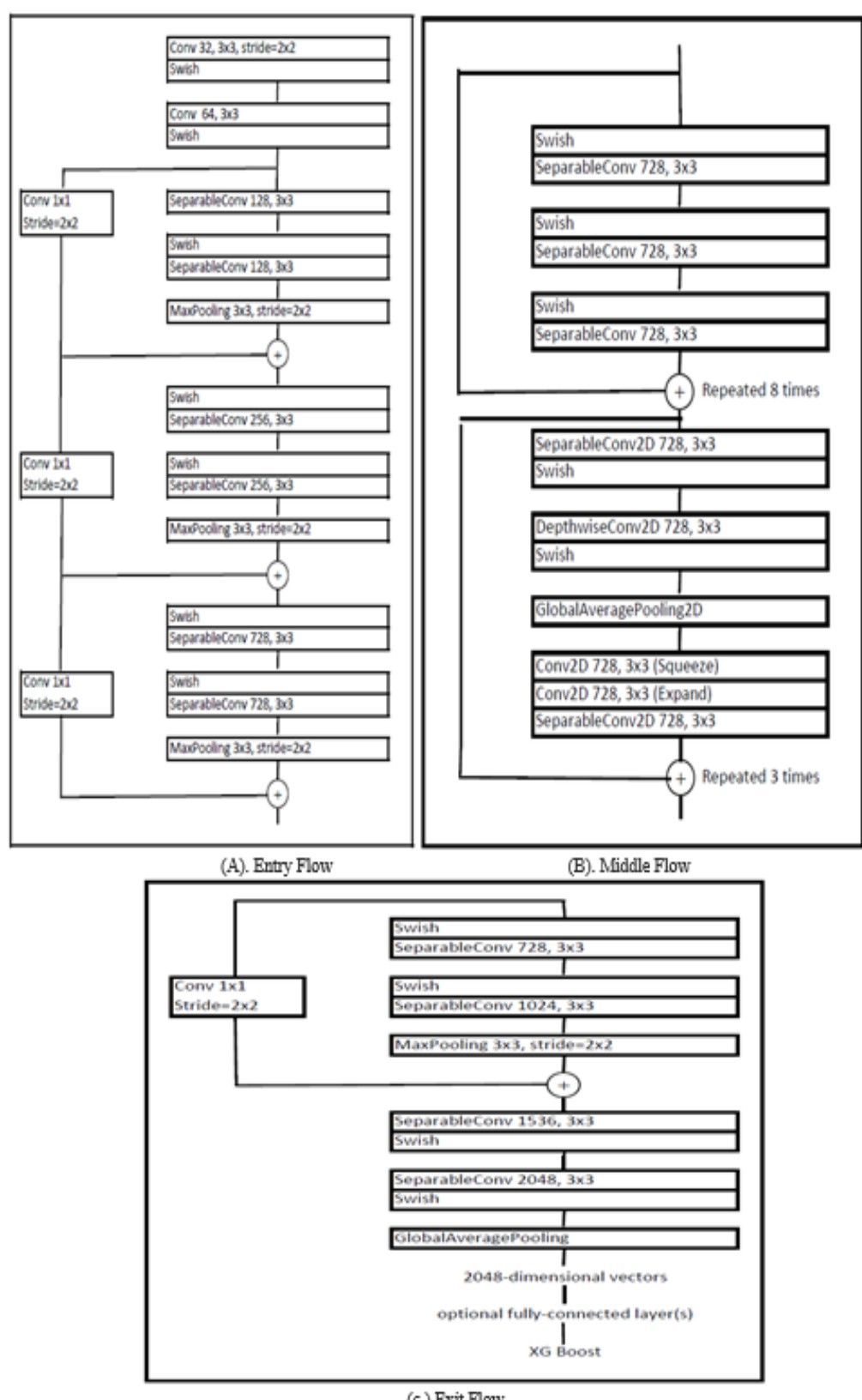

**Figure 4.** Architecture of the proposed model.

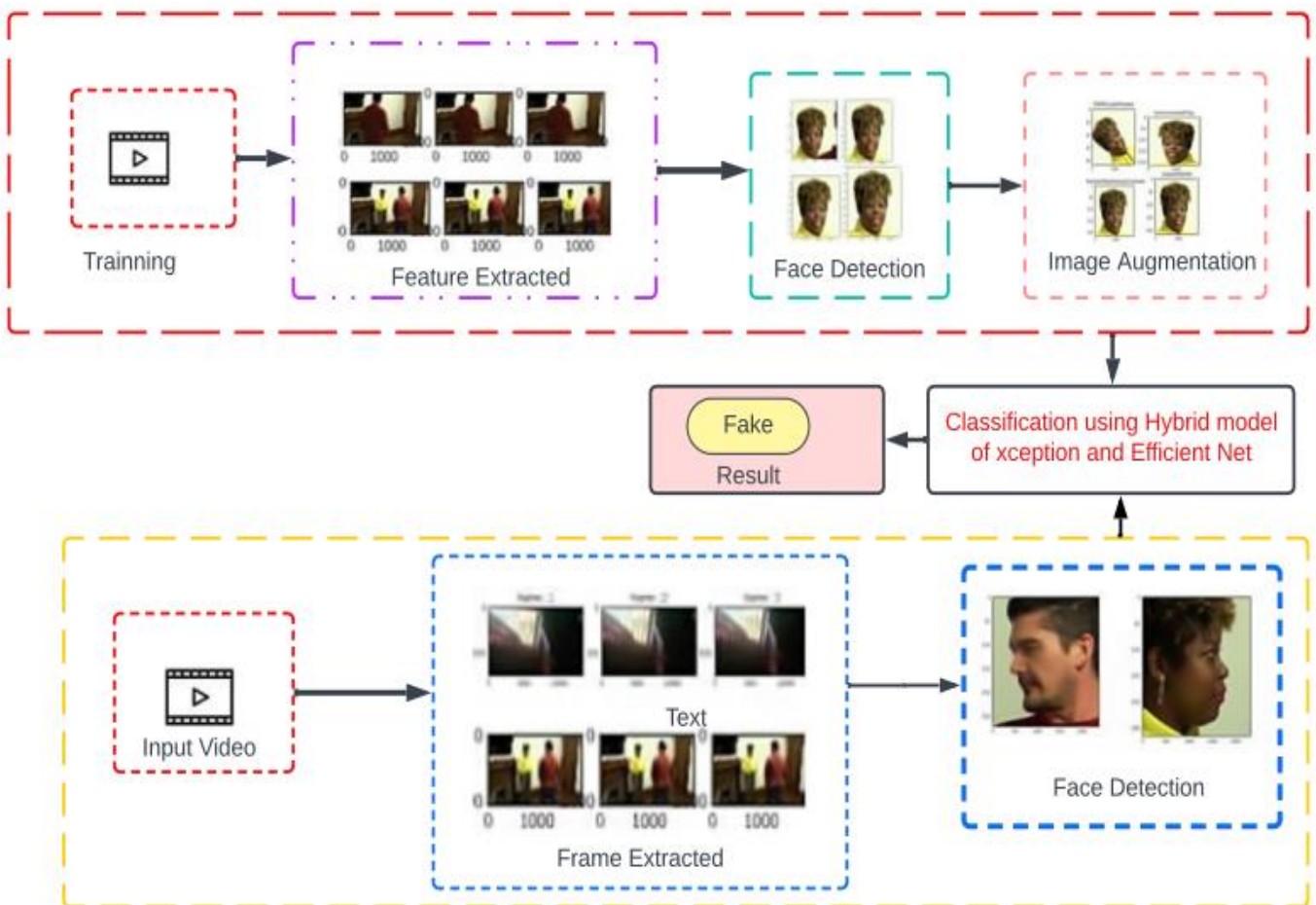

**Figure 5.** Flow chart of deep fake detection with a sample video.

## 5. Experiments

### 5.1. Dataset

The dataset plays a critical role while developing a deep learning model. Many times, results are based on the dataset selected. We selected the DFDC (DeepFake Detection Challenge) dataset [31]. Its size is approximately 470 GB. This dataset is taken from Kaggle Competition, provided by Google.

It is a generalized dataset but images are not provided for certain events like eye blinking, swapped face, lip movement, etc. As a result, the model is not able to predict the desired result. Performing any simulation using this dataset requires us to create and refine different rules to pre-process the data in order to enhance the performance of trained model.

We divided this dataset into training, validation, and testing sets. This dataset contains both real and fake videos. Videos are converted into frames and frames having faces were selected to provide input to the model shown in Figure 6. After converting all the videos into frames and detecting faces from all the frames, we used 65,234 real images and 68,258 fake images for training our model. For validating, 5876 real images and 5698 fake images were used. Testing was performed on 9785 real images and 9542 fake images.

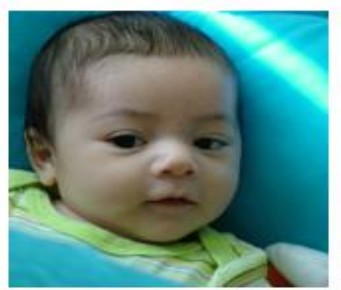
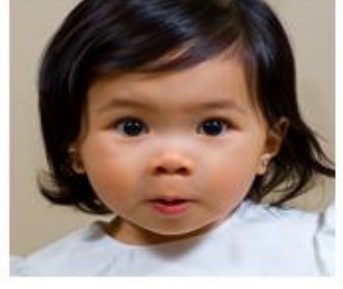
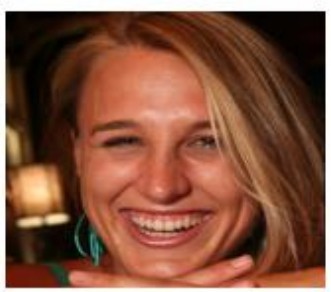
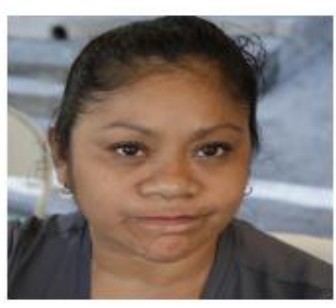

**Figure 6.** Sample images from the DFDC dataset.

The statistics of the DFDC dataset that we used for training, validation, and testing of our model are shown in Table 3, specifically the number of frames having faces. Other frames (not having faces) extracted from the videos were not used while training or testing our model. After removing the frames without faces, we found 65,234 real and 68,258 fake images for validating our model, 5876 real and 5698 fake images for training our model, and 9785 real and 9542 fake images for testing our model. We kept a large amount of data for training our model so that it learns maximum features, as well as some unseen data for testing the performance of our model.

**Table 3.** Dataset statistics.

| Dataset | | DFDC (DeepFake Detection Challenge) |
|---|---|---|
| **Size** | | **470 GB** |
| Training | Frames having faces (Real) | 65,234 |
| | Frames having faces (Fake) | 68,258 |
| Validation | Frames having faces (Real) | 5876 |
| | Frames having faces (Fake) | 5698 |
| Testing | Frames having faces (Real) | 9785 |
| | Frames having faces (Fake) | 9542 |

To visualize the dataset and understand its characteristics, for feature distribution, we used t-SNE, t-distributed stochastic neighbor embedding plot. It has been used to map multi-dimensional data (images) in two dimensions. It is a nonlinear algorithm which is very useful for exploring high dimensional data and dimensional reduction.

Figure 7 shows the visualization of our dataset by giving each data point (image) a location in a two-dimensional map. It shows the real and fake images. The value 0 is used for fake images and 1 is used to represent real images.

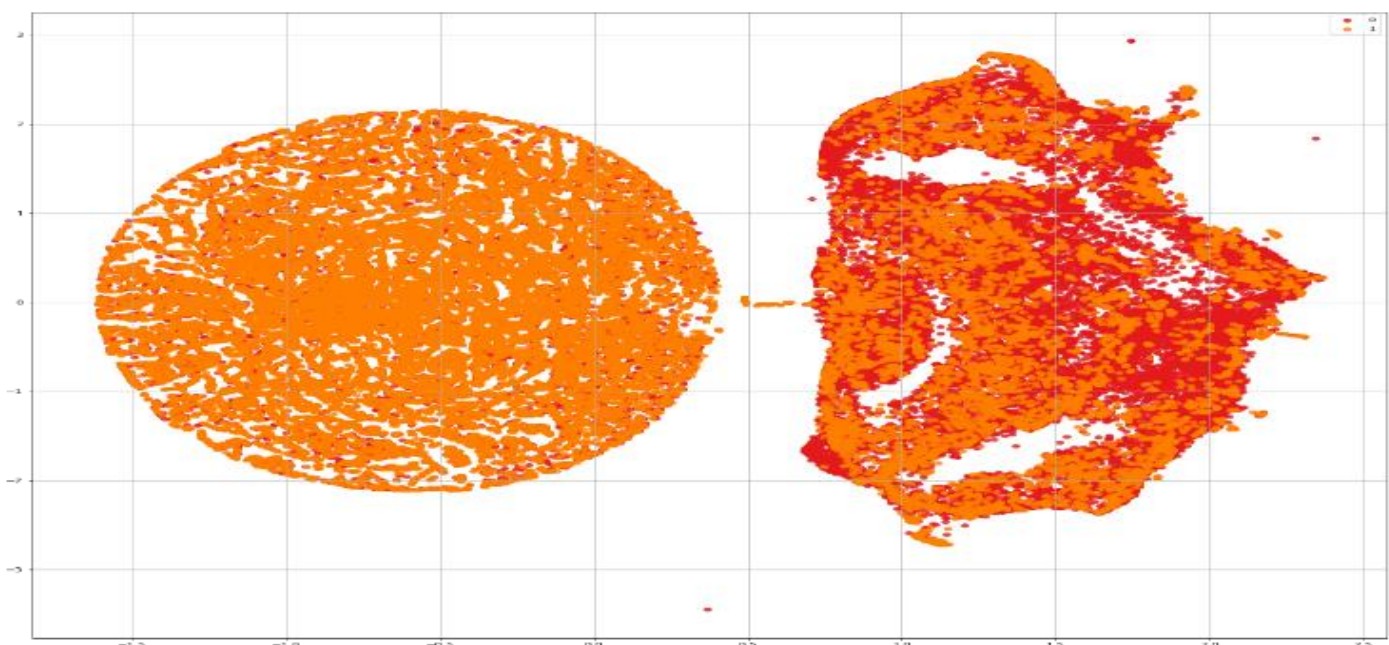

**Figure 7.** Data visualization.

### 5.2. Experimental Parameters

As the dataset used is very large, it cannot be used to train our model on any system. To create our model, to train the model, to experiment on various parameters, to select the best combinations of parameters of both the models, and to test our model, we used a system with GPU NVIDIA Tesla T4, 4 GPUs, 16 core CPU, 128 GB RAM, 200 GB SSD persistent disk, and ubuntu18.04 LTS for the operating system. These configurations were required for the fast and proper functioning of our model. We used the GCP AI notebook instance for training.

### 5.3. Experimental Settings

1. Drop Out: This parameter is used to reduce over fitting. To train all nodes equally, we drop out some neurons for a particular epoch. To do so, we randomly drop neurons in our model during training. This forces the network to share information between weights, which leads to an increase in its ability to generalize to new data. Some nodes having more weight may get turned off multiple times, and some nodes having less weight may not turn off even once. The chance is between 0–1. This determines the fraction of neurons that should be turned off from the previous layer. In our proposed model, we used a dropout value of 0.45.
2. Learning Rate: This parameter is tuned to control the change in the model after each time the weights are updated. The learning rate generally decides the time taken by the model to converge. It is the step that is taken by the model each time to move close to convergence. If the learning rate is too large, the model takes larger steps, and it will converge fast; as a result, there is the chance that the model may not converge exactly on the minimum, and the minimum may get missed. If the learning rate is too small, then the model will take a very large time to converge. The best learning rate is one that decreases as the model gets closer to the solution. We used Reduce LR on the plateau learning rate scheduler in Pytorch. It reduces the learning by itself if accuracy does not improve in 2–3 epochs. In our proposed model, we used a learning rate of 0.001.
3. Epochs: This parameter is the count of iterations. In each iteration, the model is trained on the full training dataset. A greater number of iterations leads to high-performance gains of the model to some extent, as in each iteration the weights are adjusted based on error calculation on the previous iteration. This reduces the error in

each following iteration. To maintain the trade-off between the time taken in training the model and performance gains, the number of epochs must be selected wisely. In our proposed model, we used 30 epochs.

4. Batch Size: This parameter defines the number of input samples that will be sent to the model at once for training. The complete training dataset is divided into batches, each with a size equal to that of batch size. The model is trained on one batch at a time. A large batch size trains the model fast and takes less memory, but the model has less learning, whereas too small a size may lead to large memory requirements and more training time for the model. Thus, the batch size must be selected, maintaining the tradeoff between accuracy and training time of the model. In our proposed model, we used 128 as the batch size.

5. Optimizer: We used a loss function to calculate the loss or the wrong predictions made by our model. Then we tried to reduce the loss function by tuning the hyper parameters. Optimizers tie loss function with the parameters of the model. They update the model in response to the outcome of the loss function. We used Adam Optimizer to optimize our model. It is an adaptive optimizer. It is the combination of AdaGrad and RMSProp, and hence combines the advantages of both.

6. Activation Function: Activation functions are an integral part of the artificial neural network. They help the model to learn complex patterns, and decide what will be fired as an input to the next neuron. The Swish activation function is a multiplication of linear and sigmoid activation functions. It has solved the problem of ReLu where the negative values are nullified to zero. We found in our model architecture that Swish performs better than the ReLu activation function.

$$\text{Swish}(x) = x \times \text{sigmoid}(x) \tag{4}$$

Table 4 shows various parameters used in our proposed model. We used these parameters after experimenting with our model on various other values and have concluded that our model performs best with these parameters.

**Table 4.** Parameters used in the proposed model.

| Parameters | Used Value |
| --- | --- |
| Drop Out | 0.45 |
| Learning Rate | 0.001 |
| Epochs | 30 |
| Batch Size | 128 |
| Optimizer | Adam |
| Activation Function | Swish |

*5.4. Interpretation of Our Model*

We used a saliency map for interpreting the predictions of our model. Although it is the oldest method, it is still the most useful way for interpretation of deep learning models. The saliency maps of various images extracted from the dataset are shown in Figure 8. It can be interpreted that our model is making correct predictions and is detecting perfect facial features. As our model is trained only for facial manipulation techniques, the saliency maps convey the same thing. If there is no face involved, than our model will directly ignore it, hence resulting in no saliency map. A red map shows that it is a fake image, and the area which is red in color tells us the manipulated area in the image. Using the saliency map, we also came to know that our model does not perform well on blurred images or under insufficient light conditions; the saliency map is just black for such images.

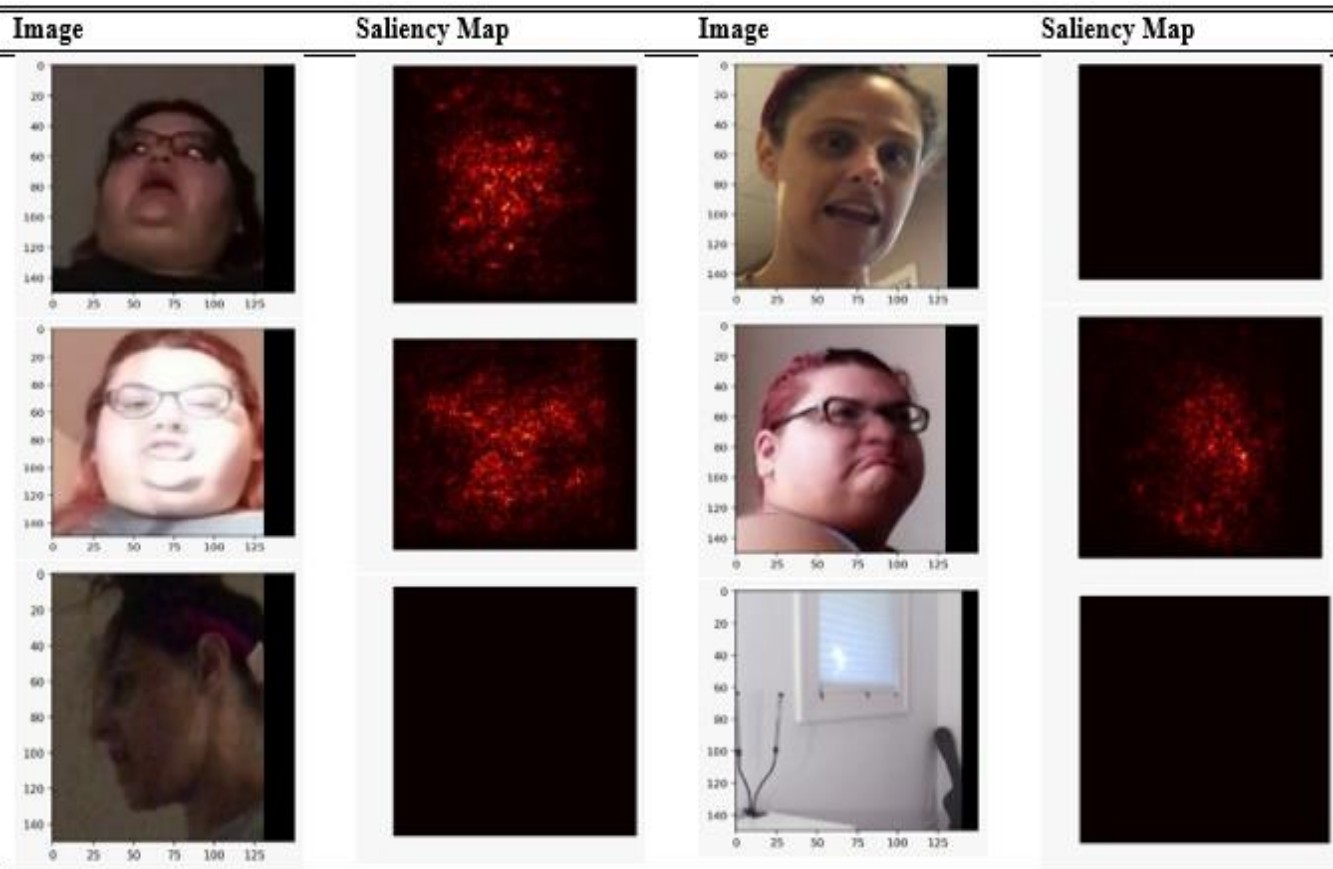

**Figure 8.** Saliency map of various images.

*5.5. Performance Evaluation*

To evaluate our model and compare it with other models, we used various evaluation metrics like Accuracy, Precision, Recall, and F1 Score. These metrics were calculated using predefined methods of the sklearn library. Mathematically these parameters can be calculated using the below equations.

Precision: The ratio of correct positive predictions to that of the total positive predictions. It shows how many correct predications our model has made out of the total positive predictions. In our case, it gives us the answer of all videos that were predicted as fake, and how many of them were actually fake, or how many of them were correctly predicted.

$$Precision = True\ Positive/(True\ Positive + False\ Positive) \tag{5}$$

Recall: The ratio of correct positive predictions to that of the total predictions of that particular class(es). It shows how many of the total positive inputs are correctly predicted by our model. In our case, it tells us about all the videos that were truly fake, and how many of them were labeled correctly by our model.

$$Recall = True\ Positive/(True\ Positive + False\ Negative) \tag{6}$$

F1 Score: The weighted average of Precision and Recall. It takes into consideration both False positives and False negatives. It is generally more useful than accuracy; however, it cannot be intuitively understood as accuracy. It is calculated as:

$$F1 - Score = 2 \times (Recall \times Precision)/(Recall + Precision) \tag{7}$$

Accuracy: The most general and intuitive term to measure performance. It is the ratio of correct predictions by the model relative to the total inputs provided to the model. It can be calculated as:

$$\text{Accuracy} = (\text{True Positive} + \text{True Negative})/(\text{True Positive} + \text{False Positive} + \text{True Negative} + \text{False Negative}) \tag{8}$$

Log Loss: In our model, the main goal is to minimize the log loss error function, as shown below. It is the most important classification metric for binary classification models based on probabilities. The lower the log loss value, the better the predictions will be and vice versa.

$$H_p(q) = \frac{-1}{N} \sum_{i=1}^{N} y_i * log(p(y_i)) + (1 - y_i) * log(1 - p(y_i)) \tag{9}$$

$y$ is the label: 1 for positive and 0 for negative.
$N$ is the number of input observations.
$p(y)$ is the probability of being positive for all $N$ inputs.
$H(q)$ is the binary cross entropy/log loss.

Confusion Matrix: A way of evaluating the performance of a classification model. It is a N $\times$ N matrix where N is the number of target classes. It is used to evaluate how well our model is performing and the types of errors our model is making. In our model, we are classifying two classes, Real and Fake. Thus, we have a 2 $\times$ 2 confusion matrix. This matrix shows the exact count of the input values that are predicted by our model for both classes. It shows the count of observations that are real and predicted-real by our model, observations that are real but predicted-fake by our model, observations that are fake and predicted-fake by our model, and observations that are fake but predicted-real by our model.

Figure 9 shows the graphical representation of the performance of our model with the increase in number of epochs. This graph shows both training as well as testing accuracy of the model on the augmented dataset. It is evident from the graph that performance was better on the training dataset; however, as we can see that the difference in performance is not large, we can say that our model does not suffers from an overfit problem. Our model is generalized and performs well on unseen data with an accuracy of 93.28%.

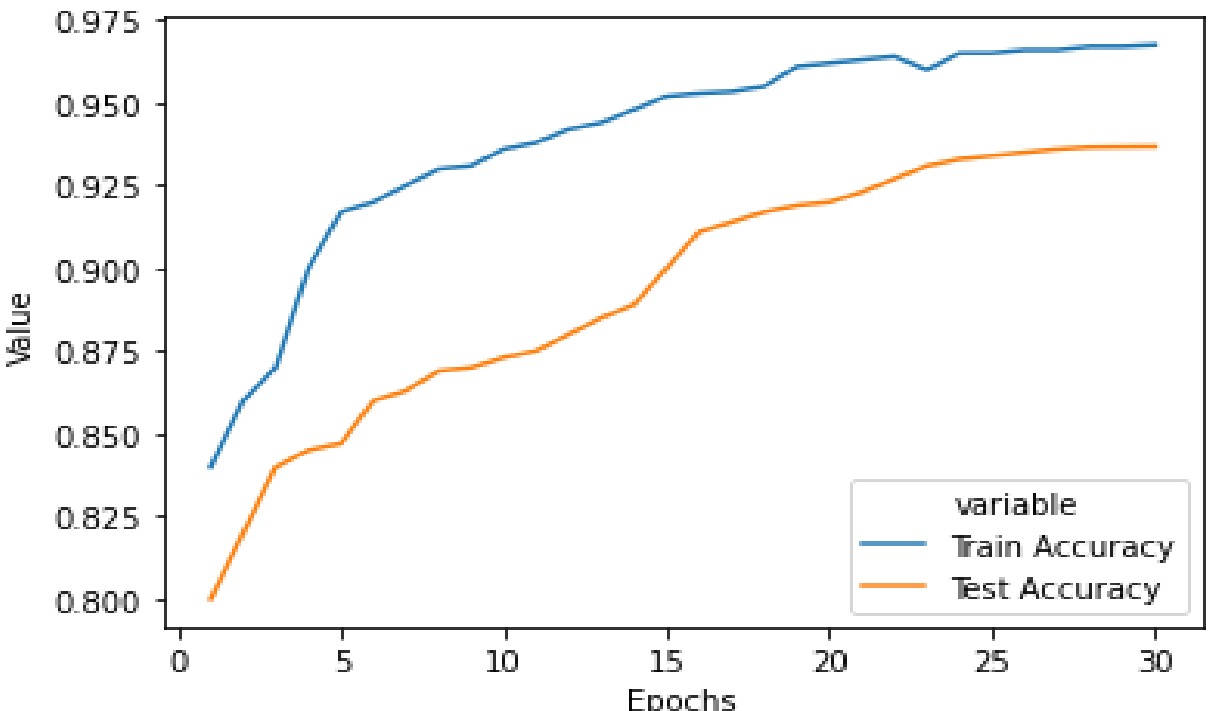

**Figure 9.** Testing and training accuracy of our proposed model on the augmented dataset.

Figure 10 shows the training and testing loss of our model on the augmented dataset. It can be easily visualized from this graph that the difference between the training loss and testing loss is small, which results in the conclusion that our model is not remembering the training data, but learning from it. After executing our model for 30 epochs, testing loss was 0.1865 whereas training loss dropped to 0.1623.

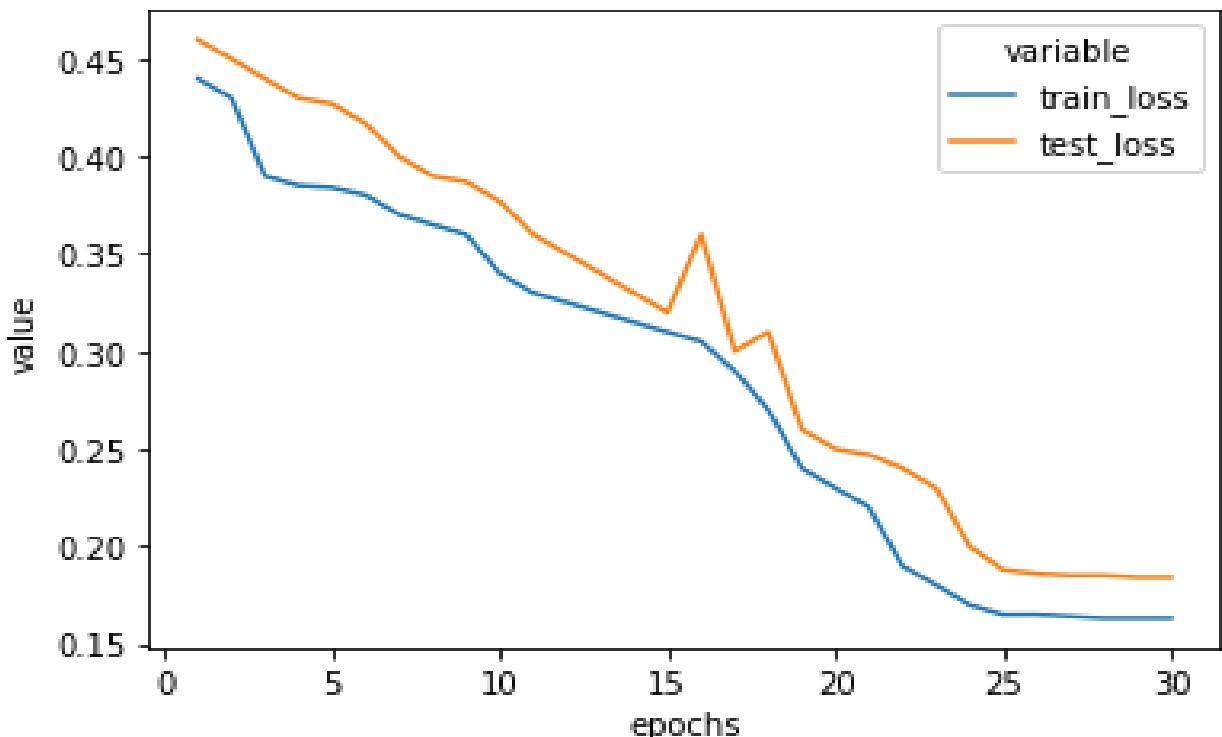

**Figure 10.** Testing loss and training loss of our proposed model on the augmented dataset.

### 5.6. Performance Evaluation and Comparison of Different Architectures

We tested the performance of Xception model architecture and Efficient Net model architecture on the augmented DFDC dataset, and evaluated the performance using a confusion matrix.

Table 5 shows the confusion matrix of the Efficient Net model. This confusion matrix was created for the unseen DFDC dataset. This matrix shows us the number of True Positive, False Positive, True Negative, and False Negative values. It can be determined from the table that out of all the real images, 8543 images were correctly classified as real images and 1242 images were misclassified as fake images. Similarly, out of all the fake images, 8448 images were correctly classified as fake images whereas 1094 images were misclassified as real images

**Table 5.** Efficient Net confusion matrix.

| N = 19,327 | Predicted Real | Predicted Fake |
| --- | --- | --- |
| Actual Real | 8543 | 1242 |
| Actual Fake | 1094 | 8448 |

Table 6 shows the confusion matrix of the Xception model. This confusion matrix was created for the unseen augmented DFDC dataset. This matrix shows us the number of True Positive, False Positive, True Negative, and False Negative values. It can be determined from the table that out of all the real images, 8792 images were correctly classified as real images and 993 images were misclassified as fake images. Similarly, out of all the fake

images, 8620 images were correctly classified as fake images whereas 922 images were misclassified as real images.

**Table 6.** Xception confusion matrix.

| N = 19,327 | Predicted Real | Predicted Fake |
|---|---|---|
| Actual Real | 8792 | 993 |
| Actual Fake | 922 | 8620 |

Table 7 shows the confusion matrix of our proposed model. This confusion matrix was created for the unseen augmented DFDC dataset. This matrix shows us the number of True Positive, False Positive, True Negative, and False Negative values. It can be determined from the table that out of all the real images, 8908 images were correctly classified as real images and 877 images were misclassified as fake images. Similarly, out of all the fake images, 8879 images were correctly classified as fake images whereas 663 images were misclassified as real images

**Table 7.** Proposed model confusion matrix.

| N = 19,327 | Predicted Real | Predicted Fake |
|---|---|---|
| Actual Real | 8908 | 877 |
| Actual Fake | 663 | 8879 |

We experimented with various face detector methods and various deep learning state of the art models combined with different classifiers. The different face detector techniques experimented with were Blaze face, YOLO [33,34], and dlib [35]. Considering that it had the most accurate and fast detection of the faces, we used Blaze face detection technique in our proposed model. Various deep learning models were used on the detected faces, including Xception [36], Efficient Net B5 [37], ResNet152 [38], InceptionResNet152V2 [17], etc., to evaluate their performance on DFDC dataset. Various classifiers like SVM [39], Logistic Regression, XGBoost [35], etc. were applied on top of these networks to classify the video as real or fake.

After evaluating and experimenting with various combinations of face detectors, deep learning networks, and classifiers, we found that BlazeFace + the Proposed Model + XGBoost performed best among all these combinations and attained the highest accuracy [40]. We compared the performance of our proposed model architecture with the Xception model architecture and Efficient Net model architecture. We proved that our proposed model architecture had the highest performance among the three. We calculated this performance using the DFDC datasets [41,42].

We measured the performance through precision, recall, F1 score, and accuracy. In addition, we considered log loss and proved that the proposed model attains minimum testing loss and the highest testing accuracy [43]. This shows that the proposed model outperforms the other models for deepfake video detection on the DFDC dataset. Table 8 shows the comparison of the proposed model with the Efficient Net and Xception models relative to Precision, Recall, F1-Score, and Accuracy.

From Table 8, it can be seen that results are poor for feature extraction using Efficient Net B5 and classifier performance using SVM. Xception and Efficient Net improved performance by up to 2%. It can also be observed that classification accuracy is low and computation time is high in a deeper network like Resnet152 [12,13]. The proposed DFN along with the YOLO model is better than Efficient Net B5. Moreover, DFN with SVM and DenseLayer achieved satisfactory performance. The highest classification accuracy of 93.28% was obtained using DFN and XGBoost [14].

**Table 8.** Performance Comparison of Proposed Method with Other Methods Using the DFDC Dataset.

| Models | Precision | Recall | F1-Score | Accuracy |
|---|---|---|---|---|
| BlazeFace + Efficient Net B5 [43] + SVM [39] | 0.7985 | 0.8094 | 0.8021 | 0.8246 |
| YOLO + Xception [36] + SVM [39] | 0.8030 | 0.7902 | 0.7853 | 0.8337 |
| MTCNN [31,41,43] + InceptionResNetV2 + XGBoost | 0.8265 | 0.8045 | 0.8218 | 0.8445 |
| YOLO + InceptionResNetV2 + XGBoost [YIX] [17] | 0.8736 | 0.8539 | 0.8636 | 0.9073 |
| YOLO + ResNet152 [38] + SVM [39] | 0.7828 | 0.8012 | 0.7878 | 0.8250 |
| YOLO + ResNet152 [38] + XGBoost | 0.8043 | 0.8129 | 0.8083 | 0.8488 |
| BlazeFace + Efficient Net B5 [43] + XGBoost | 0.8285 | 0.8145 | 0.8012 | 0.8458 |
| BlazeFace + Xception [36] + XGBoost | 0.8728 | 0.8945 | 0.8986 | 0.9037 |
| YOLO + Xception [36] + Log Reg | 0.7645 | 0.7724 | 0.7954 | 0.8102 |
| YOLO + Efficient Net B5 [43] + Log Reg | 0.7827 | 0.8152 | 0.8021 | 0.8342 |
| BlazeFace + Efficient Net B5 + XGBoost | 0.8985 | 0.9094 | 0.9021 | 0.9146 |
| YOLO + DFN+ SVM | 0.8503 | 0.8469 | 0.8517 | 0.8528 |
| YOLO + DFN + XGBoost | 0.8627 | 0.8745 | 0.8468 | 0.8762 |
| BlazeFace + DFN+ SVM | 0.8971 | 0.9069 | 0.8823 | 0.9028 |
| BlazeFace + DFN + DenseLayer [36–38] | 0.8192 | 0.8363 | 0.8241 | 0.8152 |
| BlazeFace + DFN + Log Reg | 0.9078 | 0.9186 | 0.9254 | 0.9105 |
| Proposed: BlazeFace + DFN+ XGBoost | 0.9103 | 0.9269 | 0.9217 | 0.9328 |

Figure 11 shows the comparison of the performance of our proposed model architecture graphically in terms of precision, recall, and accuracy. These results were obtained from the testing data of our dataset.

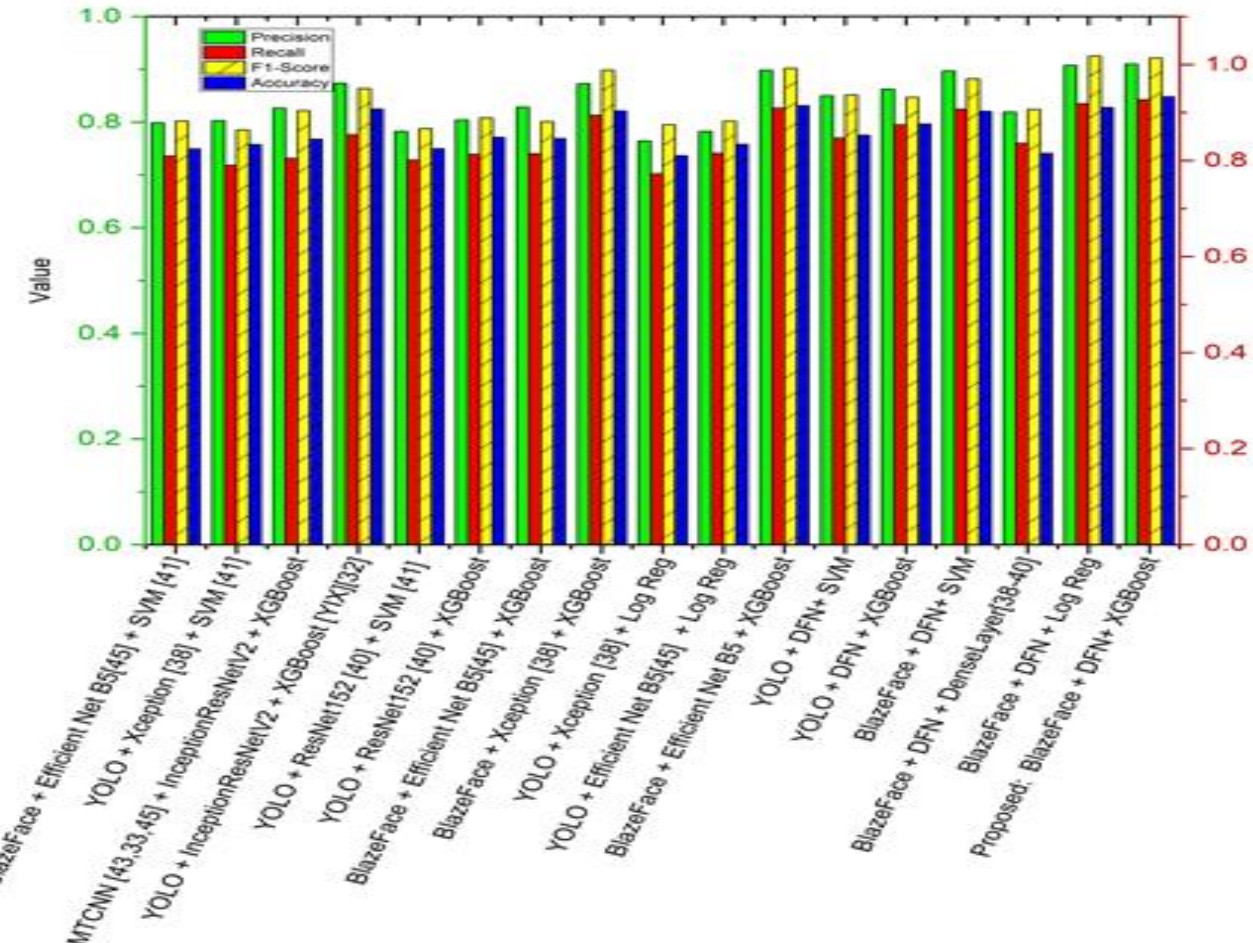

**Figure 11.** Performance measures comparison Xception and Efficient Net models with the proposed model [17,39].

Table 9 compares the performance of our proposed model in terms of training and testing loss and accuracy with the Efficient Net and Xception models [15]. These values

were obtained by training and testing all three models on the augmented data set [44,45]. We found that the Efficient Net model performs better than the Xception model in terms of both training and testing accuracy, and has less loss. Our model performs best among these models.

**Table 9.** Training and testing loss and accuracy.

| Parameters | Efficient Net | Xception | Proposed Model |
|---|---|---|---|
| Training Accuracy | 0.9562 | 0.9645 | 0.9676 |
| Testing Accuracy | 0.9086 | 0.9263 | 0.9328 |
| Training Loss | 0.1624 | 0.1345 | 0.1422 |
| Testing Loss | 0.2172 | 0.2062 | 0.1865 |

## 6. Conclusions

The quality of deepfakes is increasing rapidly, so our detection methods must improve faster to catch up. The competition between creators and catchers of deepfakes is increasing day by day. The motivation behind the proposed model for deepfake detection is that deep learning techniques themselves can solve the problems created by deep learning techniques. Several pieces of research on this have been conducted, but they are less efficient, and computation time is high. In this study, we created a new deep CNN model, DFN, that contains some building blocks of Xception and Efficient Net to reduce complexity and improve performance gains. The proposed model achieved up to 2% accuracy gain and 1.18% precision improvement compared to Efficient Net B5. In addition, DFN is fast and converges in low time compared to other state-of-the-art models. This model can also be used in other forgery detection tools currently used by cyber security experts.

In the future, a different model architecture should be designed which considers both audio and visual features for better detection of deepfakes. The model developed herein for deepfake detection can become the baseline model for other researchers in the future. Audio features were not yet taken into consideration due to noise and various other issues, and these features can help further improve the accuracy of these detection methods. Other datasets, including audio forgery, can be used to train models in different aspects. Various speech alteration methods can be combined with visual alteration methods to better explore this area of research.

Cybersecurity professionals can also work on various types of adversarial attacks possible on this model and can suggest different ways to prevent them. Machine learning models are prone to adversarial attacks, so methods can be implemented to prevent various types of attacks by taking this model as a baseline.

**Author Contributions:** N.B. Conceptualization, software, T.A.: methodology, validation, D.P.Y.: formal analysis, investigation, K.U.S.: data curation, writing—original draft preparation G.K.V.: writing—review and editing, A.K.: visualization and T.S.: supervision. All authors have read and agreed to the published version of the manuscript.

**Funding:** This research received no external funding.

**Data Availability Statement:** Not applicable.

**Conflicts of Interest:** The authors declare no conflict of interest.

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
