# Peer review of "Real-Time Advanced Computational Intelligence for Deep Fake Video Detection"

_applsci, doi:10.3390/app13053095_

Round 1

Reviewer 1 Report

Overall the paper presents the work fairly well - it explains the method. parameters and evaluation metrics clearly. Overall it is fairly easy to follow.

Although I do think that the Y axis on Figure 11 does exaggerate the difference that is truly present. There is an improvement there, but I think this needs to be highlighted as being more minor than it currently is within the text. 

Figure 2 does not appear to accurately demonstrate each of these types for the images shown. Horizontal flip for example on the first row -  which on the second row it has done, but it also has had additional brightening of the image. 

Page 10 needs to be fixed.

Page 11 is too blurry, please fix.

Author Response

Reviewer Comment 1

QA1. Overall the paper presents the work fairly well - it explains the method. parameters and evaluation metrics clearly. Overall it is fairly easy to follow.

Response:  Thanks for your valuable time and effort. 

QA2. Although I do think that the Y axis on Figure 11 does exaggerate the difference that is truly present. There is an improvement there, but I think this needs to be highlighted as being more minor than it currently is within the text. 

Response:  We have redrawn the figure and it is now more visible.

QA3. Figure 2 does not appear to accurately demonstrate each of these types for the images shown. Horizontal flip for example on the first row -  which on the second row it has done, but it also has had additional brightening of the image. 

 Response:  We have updated the figure 2 as per your comments.

QA4. Page 10 needs to be fixed.

Response:  Thanks for your valuable comments, we have updated the formatting of page 10.

QA5. Page 11 is too blurry, please fix.

Response: Thanks for your valuable comment, we have updated the page 11.

Reviewer 2 Report

I commend the authors for writing this article in a straightforward and concise way that is easy to understand and comprehend.

Lines 64 – 67 invoked a statistic finding as evidence; however, there is no citation provided.

The literature review section will be more readable if a tabular summary of the existing techniques alongside their current model metrics is provided.

Please provide the actual yields in lines 426-433 instead of author interpretations of the metrics.

Lines 441 and 442, what then is the actual, acceptable dataset count to train a model, and by what or whose standards?

Author Response

Reviewer Comment 2

I commend the authors for writing this article in a straightforward and concise way that is easy to understand and comprehend.

QA1: Lines 64 – 67 invoked a statistic finding as evidence; however, there is no citation provided.

Response:  Thanks for your valuable comment, we have updated the line 64-67 as per your comments.

QA2: The literature review section will be more readable if a tabular summary of the existing techniques alongside their current model metrics is provided.

Response: We have updated the manuscript and add a comparative analysis of exiting techniques alongside their current model metrics is provided in fig 11 and table 8.

QA3: Please provide the actual yields in lines 426-433 instead of author interpretations of the metrics.

 Response: We have updated the lines 426-433 and improve the fluency of these line.

QA4: Lines 441 and 442, what then is the actual, acceptable dataset count to train a model, and by what or whose standards?

Response: We have updated the Lines 441 and 442 and improve the fluency of these line.
